# Non-Linear Modeling of Motor Development in Typically Developing Children and Youth Aged 5–18 Years Using Robot-Based Behavioral Assessments

**DOI:** 10.3390/bioengineering12111240

**Published:** 2025-11-12

**Authors:** Stephan C. D. Dobri, Stephen H. Scott, T. Claire Davies

**Affiliations:** 1Department of Mechanical and Materials Engineering, Queen’s University, Kingston, ON K7L 3N6, Canada; claire.davies@queensu.ca; 2Department of Biomedical and Molecular Sciences, Queen’s University, Kingston, ON K7L 3N6, Canada; steve.scott@queensu.ca

**Keywords:** motor development, robotic assessment, simulations, non-linear, children, youth, adolescent, motor function, reaction time

## Abstract

Clinical tasks are often used to differentiate the motor performance of individuals who have impaired function. However, these are not as accurate and repeatable as robotic tasks. Additionally, motor development occurs rapidly at early ages and slows as they reach adulthood, resulting in a non-linear model of performance. There is also evidence that variability in performance changes as children and youth age. Accurate normative models of performance are necessary to identify deficiencies in motor performance and to track the efficacy of therapies. This work aimed to create normative models of motor development based on robotic assessments in typically developing children and youth. Two hundred and eighty-eight participants who are typically developing (ages 5–18) completed a robotic point-to-point reaching task and an object-hitting task using the Kinarm Exoskeleton. Exponential or quadratic curves were fit to performance parameters generated by Kinarm to model typical performance. These models included a linear term to account for changing variabilities with age. Most performance parameters showed improvement with age, and none showed deterioration. Some parameters showed large changes in variability in performance with age, with up to a 74% decrease in the range of typical performance. Reduced variability occurs with age, indicating the need to account for differences in variability when developing models of typical motor performance in children and youth. The models that are used to identify deficits in motor performance should account for changing variability in data and changing repeatability with age to increase the accuracy of identification of deficits.

## 1. Introduction

Differentiating between normal and abnormal capabilities in children requires an understanding of how basic neurological functions change with increasing age. The present study was conducted to generate a set of normative data that can be used for comparisons with clinical populations in quantifying how upper limb visually guided reaching and object-hit tasks change with age.

Child development has conventionally been categorized through different milestones that correspond to improvements in motor, cognitive, or overall function [1,2,3,4,5,6,7,8]. “Near-adult” performance is defined through these milestones; once a child has hit the final milestone, they are considered to have achieved “near-adult” performance [8]. Different aspects of motor function and coordination reach these “near-adult” levels at different ages. For example, proprioception tends to reach “near-adult” performance near 8 years of age, whereas other aspects of coordination do not mature until 12–14 years of age [8,9,10,11]. One can measure different aspects of function relative to age to create developmental curves. To evaluate whether a child meets the typical milestones of motor performance, their data can be plotted against chronological age to allow comparisons against developmental expectations of motor function. Any measure that lies beyond the typical normative models can allow clinicians to identify deficits [12].

Typically, these curves are based on clinical assessments. For example, the Purdue Pegboard and Bruininks–Oseretsky Test (BOT) of Motor Proficiency have been validated on large populations of typically developing children to determine normative performance intervals [13,14]. The PPB gives aggregate measures of function (the number of pegs placed and assemblies built), which is also affected by manual dexterity, and can identify impairment, but not what part of the movement is impaired. The BOT is used as a discriminative and evaluative measure to characterize motor performance; however, there is weak test/retest reliability, which limits therapist confidence [15].

Robotic tasks have been identified as a promising way to measure different aspects of function and deliver therapies for patients with different injuries or disabilities [16]. Robotic tasks offer highly repeatable, objective, and accurate measures of function with high inter-rater reliability [16,17]. The measures from robotic tasks are objective and accurate, making them ideal for creating developmental curves. Clinical assessments are typically coarse measures, while robotic measures are quantifiable and easily detectable.

One of the robotic devices often used for these assessments is the Kinarm Exoskeleton Robot Lab (Kinarm, Kingston, ON, Canada) [18]. The robot has been used in studies with adult participants to quantify different aspects of motor function, cognitive function, and proprioception [19,20,21,22,23]. In adult populations, robotic devices have been used extensively to quantify typical age-related changes, as well as deficits caused by injuries or illnesses [19,20,21,22,23]. Additionally, robots have been used for rehabilitation in adults, often for those who have experienced a stroke [19].

Robotic assessments of motor function have been shown to be highly objective and repeatable; however, normative models of child development from robotic assessments are missing from the the current literature [17,24]. Researchers have used robotic assessments to make group comparisons between typically developing populations and clinical populations, such as children with cerebral palsy [24], but a normative database is not available. A systematic review [24] found 19 studies, of which only eight included more than 50 participants.

Additionally, previous research that created models of motor development has not accounted for changing variabilities in performance across childhood and adolescence [24]. Changing variabilities in motor performance have been reported across childhood and adolescence [11,25,26]. It is important to include age-related changes in the variability of motor performance in normative models. Including this variability would allow models to better represent expected behavior and identify deficits in performance. While other groups, for example, Grohs et al. and Hawe et al. [27,28], have developed normative models of typically developing performance for the VGR and OH tasks using Kinarm [27,28], their numbers have been limited to 155 and 146 participants, respectively, and they assumed constant variability in performance across the age group. While each of these studies included participants from 6–19 years, only the mean and standard deviation of the demographics for each of the groups were reported, so it is unknown if the data were biased towards a specific age.

The purpose of this study was to measure the performance in a cohort of typically developing participants performing two robot-based motor tasks and to create normative models to quantify motor development. The tasks used for analysis included Visually Guided Reaching (VGR) and Object Hit (OH). These tasks were chosen as they test basic unimanual reaching and bimanual motor skills. VGR assessed basic unimanual reaching performance, which has been shown to be impaired in participants with cerebral palsy and developmental coordination disorder [27,29,30]. OH was assessed to quantify bimanual motor function and was previously used to compare the performance of typically developing children and those with hemiparetic cerebral palsy [28,31]. The normative models of typical development can be used to identify deficits in performance in various patient populations and to track the efficacy of therapies.

## 2. Materials and Methods

The cross-sectional study was approved by the Health Sciences and Affiliated Hospitals Research Ethics Board (HSREB) at Queen’s University, Kingston, Ontario, Canada (application number 6004951, approved 9 May 2016) in accordance with the Helsinki Declaration of 1975, as revised in 2000 and 2008.

### 2.1. Participants

Typically developing children and youth were recruited from the Kingston, Ontario, Canada area. The participants were between 5–18 years old in chronological age. Typical development is characterized by basic tenets, including the following: gross motor development progresses in a cephalo-caudal sequence; fine motor development progresses from a midline to lateral sequence; primary motor patterns are integrated into more complex patterns; the sequence of developmental skills remains the same—though the rate may differ; and progression from generalized reflexive responses to a more specific and purposeful pattern [32]. Participants were recruited through summer camps run by the university, word of mouth, and online flyers. Data were collected from June 2016–March 2020. Signed guardian consent and participant assent were obtained prior to testing. Participants were excluded if they had documented evidence of recent concussion, a history of physical, intellectual, or neurological disorders, or had perceptual impairments such as visual or auditory deficits.

### 2.2. Robotic Apparatus and Tasks

The Kinarm Exoskeleton Lab (Kinarm, Kingston, ON, Canada) was used in this study [18]. Participants sat in a system-integrated chair with their arms supported against gravity by the exoskeleton. The robot allows free planar movement of the upper limbs while recording elbow and shoulder joint kinematics. Hand position and kinematics were calculated from the joint kinematics. Participants sat in front of a virtual reality display, with vision of their hands and arms occluded. Hand position feedback was displayed in each task on the virtual reality display aligned with the horizontal workspace.

The Visually Guided Reaching (VGR) task assesses goal-directed reaching [23]. There are four peripheral targets set in a square, 6 cm apart, with a fifth starting target in the center of the square. Participants are instructed to reach as quickly and accurately as possible to each target as they appear. The peripheral targets appear in a pseudo-random order, six times each for a total of 24 reaches per arm. Each reach consists of moving out to the peripheral target, then back to the central target when it reappears. Participants complete the task once with their dominant and non-dominant hands. This task is represented in Figure 1A. Motor performance was quantified using ten parameters [23]:Posture Speed: The median hand speed (in m/s) across all trials while the hand should be at rest.Reaction Time (RT): The time (in seconds) from when the peripheral target appears to when hand movement onset occurs.Initial Direction Angle: The angular deviation between (a) a straight line from the hand position at movement onset to the hand position after the initial movement phase, and (b) a straight line between the hand position at movement onset to the target. The absolute value of this deviation is calculated and used for each trial, and the median value from all trials is reported.Initial Distance Ratio: The ratio of (a) the distance moved during the first movement phase and (b) the total distance travelled by the hand throughout the movement. The median value of all trials is reported.Initial Speed Ratio: The ratio of (a) the maximum speed of the hand during the initial movement phase and (b) the maximum hand speed during the movement. The mean value of all trials is reported.Speed Maxima Count: The number of speed maxima throughout the movement. The mean value of all trials is reported.Movement Time: The time elapsed throughout the entire movement. The median value of all trials is reported.Path Length Ratio (PLR): The ratio of the hand path traveled between movement onset and offset, and the shortest path between the two targets.Max Speed: the maximum hand speed (in m/s) during the reaching movement. The median value of all trials is reported.Min–Max Speed Difference (MMSD): The difference between speed maxima and minima throughout the movement.

The Object Hit (OH) task measures bimanual coordination [18,33]. The task consists of red balls moving towards the participant in the horizontal workspace. Green paddles are aligned with the participant’s index fingers (Figure 1B). The participant is instructed to hit the balls away with the green paddles. The robotic apparatus provides haptic and visual feedback when the ball is hit. The number of balls and the speed at which they fall increase throughout the task (just over two minutes total time), increasing the difficulty as the task progresses. The task was assessed with 12 parameters [18,23]:Total Hits (TH): The total number of balls the participant hit with both hands of the 300 dropped down the screen.Median Error: The percentage of the way through the task (based on the number of balls dropped) when half of the misses have occurred. For example, if a participant misses 60 balls during the task, the median error is the ball number at which the 30th ball is missed divided by 300 balls.Miss Bias: A quantification of a bias in misses toward one side of the workspace to the other (left-to-right only, reported in cm). The number of misses in each bin is counted, and a weighted mean is calculated:
(1)$$weighted\;mean=\frac{\Sigma_{i=1}^{10}x_{i}\times w_{m,i}}{\Sigma_{i=1}^{10}w_{m,i}}$$
where *x_i_* is the *x*-position of the *ith* bin, and *w_m,i_* is the number of misses in the *ith* bin.Hand Bias of Hits (HBH): A quantification of the bias in hits between the hands, calculated as [(Hits Dominant − Hits Non-Dominant)/Total Hits].Hits Dominant: The number of targets hit by a participant’s dominant hand.Hits Non-Dominant: The number of targets hit by a participant’s non-dominant hand.Hand Speed Dominant: The mean hand speed of a participant’s dominant hand (in m/s) during the task.Hand Speed Non-Dominant: The mean hand speed of a participant’s non-dominant hand (in m/s) during the task.Hand Speed Bias: A quantification of the bias in hand speed between the hands, calculated as [(Hand Speed Dominant − Hand Speed Non-Dominant)/(Hand Speed Dominant + Hand Speed Non-Dominant)].Movement Area Dominant: (Dominant or Non-Dominant): The area of the workspace (in m^2^) covered by the participant’s dominant hand during the task. It is determined by the convex hull (polygon) that includes the boundaries of the movement trajectory of the hand in question.Movement Area Non-Dominant: The area of the workspace (in m^2^) covered by the participant’s non-dominant hand during the task.Movement Area Bias (MAB): A quantification of the bias in the movement area between hands, calculated as: [(Movement Area Dominant − Movement Area Non-Dominant)/(Movement Area Dominant + Movement Area Non-Dominant)].

### 2.3. Data Analysis

Data for each parameter from both tasks were fit with one of two curves, exponential or quadratic, as shown below:(2)$$y=a_{0}\times e^{a_{1}\times age}+a_{2}\times sex+a_{3}\times hand+a_{4}$$(3)$$y=a_{0}\times age^{2}+a_{1}\times age+a_{2}\times sex+a_{3}\times hand+a_{4}$$

The variables “*sex*” and “*hand*” are dummy-coded variables (1 or 0) representing the participant’s sex and the hand (dominant or non-dominant) that completed the task. Since handedness was represented by a dummy-coded variable, data from both hands for all participants were used to generate normative ranges for VGR parameters, effectively doubling the dataset size to 576 data points for that method. The “*hand*” variable was not included in the model for the parameters from the OH task, as both hands were used during the task. The model does not allow for comparisons between right-hand versus left-hand dominance.

Curves were fit using the MATLAB function “nlinfit.m” (Mathwork, Inc., Natick, MA, USA, version 2021b). An exponential fit was used for parameters with only positive values. Exponential curves require a good initial estimate of the coefficients for the curve-fitting function to produce accurate results [34]. Initial estimates were calculated for the coefficients *a*_0_, *a*_1_, and *a*_4_ in the exponential equation using a method described by Foss [34]. By taking the natural logarithm of the values and fitting a line, the coefficients of the line were used as initial estimates for the exponential curve fitting. This method does not work with data that has negative or zero values, as the logarithmic value would result in imaginary and infinite values. Thus, quadratic curves were fit to any parameter that included both positive and negative values.

Ten-fold cross-validation was used to avoid over-fitting models to the data. The data were divided into ten equal groups; nine groups were used to train the model, and the tenth group was used to test the model. The fitting was completed ten times, with each group being used once as the testing dataset. The model that performed best (according to the r-squared value of the test dataset) was then used as the model of the data.

Next, a linear model of the absolute value of the residuals relative to age was fit using the same ten-fold cross-validation technique. The linear model accounted for age-related changes in the variability of the data. The model of residuals was used to calculate z-scores, as described by Altman [35]. An additional scaling factor was included to generate an unbiased measure of absolute deviation. The equations used to generate the z-scores are shown below:(4)$$s=\sqrt{\frac{\pi }{2}\times \frac{n}{n-DOF}}\times model\;of\;residuals$$(5)$$z=\frac{Residuals}{s}$$
where *DOF* is the degrees of freedom of the model, and *n* is the number of data points.

Three rounds of outlier removal were conducted based on the calculated z-scores. The mean and standard deviation were calculated, and any data points outside three standard deviations above or below the mean were identified as outliers. The mean and standard deviation were then recalculated, excluding the outliers, and any new outliers were again identified. This process was completed to ensure that large outliers did not affect the subsequent testing for normality of the distribution of z-scores.

The z-scores were then tested for normality using the Shapiro–Wilks test, and the skew and kurtosis were calculated. If the z-scores failed the Shapiro–Wilks test but the absolute value of the skew was less than 0.6, and the kurtosis was between 2.4 and 3.6, the z-scores were considered “normal enough”, as outlined by Pearson and Please [36]. If the z-scores failed the Shapiro–Wilks test and the skew or kurtosis were outside the acceptable ranges, the data were transformed using the following:(6)$$Logarithmic\;Transform:y=\log\left(values-\min\left(values\right)+1\right)$$(7)$$Square-root\;Transform:y=\sqrt{values-\min\left(values\right)+1}$$(8)$$Inverse\;Transform:y=\frac{1}{values-\min\left(values\right)+1}$$

To ensure that the transforms calculated only real, finite numbers, each data point was shifted by subtracting the minimum value in the dataset and adding one. For example, if the minimum value of a parameter was three, each data point for that parameter would have the value 3 subtracted and then the value 1 added. The shifting ensured the new minimum value was one, and all values were positive. The three transforms were applied separately to the data, and the same curve fitting previously described was completed for the transformed data. The best transform was the one that resulted in normally distributed z-scores. If multiple transforms produced normal z-scores, the transform with the highest r-squared value and a normal distribution was chosen. If all three transforms failed to produce normally distributed z-scores, the models from the untransformed data were kept for simplicity. The z-scores with a normal distribution were then used to create ranges of typical performance.

Figure 2 below shows a schematic of how the normative models were created.

## 3. Results

Two hundred and eighty-eight typically developing children and youth participated in the study. The participant demographics are highlighted in Figure 3 below. The mean age of the participants was 13 years, with a standard deviation of 3.2 years. Nearly twice as many participants were male as were female. Most participants were age 11 or older.

One can observe age-related differences in motor function from the visual representations of performance parameters of the reaching trials of the VGR task. Figure 4 compares the hand path trajectories and hand speed recordings from a participant in the youngest age group (5–10 years old) and a participant from the oldest age group (13–18 years old). Both participants were matched by sex and hand dominance, and the data shown display reaches with the dominant hand. Individual lines represent separate reaching trials. Figure 4A shows the hand path trajectories for reaching out and back. The younger participant (top) had more inter-reach variability than the older participant, as demonstrated by the wider spread of individual lines. They also had more corrective movements within each reach, as shown by the less-straight movements and overshooting of the targets (represented as black circles). These reaching behaviors were also quantified with the Path Length Ratio and Min–Max Speed Difference.

Figure 4B shows hand speed profiles for different reaches. The younger participant had a slower reaction time, as seen by the later movement onset, and the speed profiles again show more corrective movements from the multiple speed peaks.

Curves fit to six of the performance parameters using the presented curve-fitting algorithm are shown in Figure 5. The error bars were generated using the results from the simulations from uniform datasets discussed previously by Dobri et al. [37]. Dobri et al. compared the performance of the curve fitting between uniformly distributed datasets and the dataset collected for this analysis. They found that the exponential curves fit similarly well between this dataset and uniform datasets of a similar size for ages 9 years and above [37]. The curve fitting was less reliable for ages 8 and below. Ten more participants at each age, 5–7 years, would need to be assessed to achieve the same level of confidence in the curve fitting at 5 years as the current model has at 9 years.

Figure 5A–C show three performance parameters from the VGR task, whereas Figure 5D–F show three parameters from the OH task. Both RT and PLR (Figure 5A,B) show a decaying exponential relationship with age and a decreasing performance interval width. The typical performance interval widths (i.e., 95% band in shaded red) for RT decreased from 0.2680 s at age 5 to 0.0685 s at age 18, which reflects a 74% decrease in the variability in typical performance. Interval widths decreased from 0.6874 to 0.1867 for PLR (73% decrease). MMSD showed a linear relation with age, TH showed an increasing exponential relation, and HBH and MAB showed quadratic relations.

The performance parameters from VGR quantify some of the qualitative aspects of reaching movements noted in the participants in Figure 4. The hand path traces highlight that the younger participant deviated from a straight path more than the older participant. The deviation would result in a larger PLR, which would decrease as participants age, as is seen in Figure 4. As noted in the speed plots in Figure 4, the reaction time of the older participant is shorter than the younger participant, and this is also reflected in RT in Figure 5.

Curves were also fit to the remaining 16 parameters from both the VGR and OH tasks. These models can be found in the Appendix A.

The percentage difference in the mean value of the fitted curves across the age range was also calculated, and is shown in Table 1. Most performance parameters showed improvement with age, as demonstrated by the percentage change in the mean value of the fitted curve across the age range. For example, the reaction time of an 18-year-old was 54% faster than that of a 5-year-old, as shown by the 54% decrease in reaction time. The 18-year-old participants also showed a 25% increase in max speed, with a 31% decrease in MMSD. The increase in speed and decrease in MMSD indicate that participants were making faster, smoother reaches as age increased. Smoother reaches are also demonstrated in Figure 3 and PLR.

Many performance parameters showed changes in the width of the 95% performance confidence intervals, indicating a change in variability in performance with age. For example, hand speed with both the dominant and non-dominant hands in the OH task became much more consistent among older participants than younger participants.

The dummy-coded variables for participant sex and hand did show that there were differences in performance based on these variables. These differences were generally small, but not zero. Since these variables cause a constant shift in the curve, the impact of these differences would change across the age range. For example, male participants tended to hit eight balls more during the OH task than female participants. At age 5, if a participant hit only 95 balls, that is an 8% difference expected based on sex. At age 18, when a participant was expected to hit closer to 225 balls, that is only a 4% difference. The dummy-coded variables for sex and participant handedness can be found in the tables in the Supplemental Material.

## 4. Discussion

This dataset of typically developing participants was nearly twice the size of the next largest study of robotic tasks undertaken by groups of typically developing children and youth (288 participants compared to 155) [24,27,28,31]. Grohs et al. and Hawe et al. [27,28] developed normative models of typically developing performance for the VGR and OH tasks using Kinarm [27,28] to evaluate differences in performance compared to participants with hemiparetic cerebral palsy and developmental coordination disorder. The OH task performed by that group included a slightly different Kinarm environment, as the workspace was smaller than the version used here. The VGR task was the same between the two groups. For that study, comparisons were made between the parameters: Total Hits (TH), Reaction Time (RT), and Path Length Ratio (PLR). Curves that were quadratic with age were fit to data, which were separated between dominant and non-dominant hand performance before generating curves, and constant widths were used for the normative performance ranges [27,28]. The mean and standard deviation of the age of the participants of both datasets were similar, although the standard deviation of the current sample was smaller (3.2 years vs. 4.5 years in the previous studies). While the distribution across the age range for the smaller dataset is unknown, the larger standard deviation indicates the ages are more evenly spread than in the current study.

Collecting a sufficiently large and representative dataset poses logistical and practical challenges, but it is crucial for creating normative models of development. These robotic assessments require access to specialized equipment (the Kinarm Exoskeleton) and can take significant amounts of time to assess each participant. Performing a single assessment with the two Kinarm tasks in this study may take up to 25 min in total, depending on the experience of the operators to set up the participant and the abilities of the participant. Additionally, Kingston, ON, Canada, is a relatively small community with a population of 132,000, which can make it more difficult to recruit a sufficient number of participants in a practical timeframe. Additionally, data collection occurred in hospital settings, with limited public transit access. The location of data collection further reduced the potential recruitment pool to participants who had access to private transit or were located near limited public transit systems.

Kinarm had originally designed the Exoskeleton for use with adult participants, so the tasks needed to be modified for smaller participants. Since these modifications were not standardized across research groups, combining datasets is a challenge. Kinarm now supplies modified tasks for child participants, making future collaborations possible. Collecting a sufficiently large and representative sample of participants is critical for developing normative models of motor development. Without collecting a representative sample across all ages, models do not accurately reflect the general population. While this may be acceptable in exploratory work, normative models of development must be created from representative samples or have limited value in clinical settings.

A novel feature of our normative models is that they account for changing variability with age [24]. There are many biological factors that cause variability to decrease as children mature, including neural system maturation and improved feedforward regulation [8,11,26,38]. Capturing these changes is critical for accurate identification of motor deficits across the age range. The widths of the performance ranges calculated for this manuscript increased from 82 to 87 hits for TH (6% increase), and decreased from 0.268 to 0.083 s for RT (74%) and 0.693 to 0.210 for PLR (75%) as the individual nears adulthood. The fixed-width ranges found in the Grohs et al. and Hawe et al. studies were 82 hits for TH, 0.2 s for RT, and 0.4 for PLR [27,28]. The fixed values are within the changing interval ranges for all three of these parameters; however, the differences in the ranges for RT and PLR are important. The range calculated by the algorithm in the current study for RT is 34% larger at the younger ages than the fixed-width range and 59% smaller at the older ages. Similarly, for PLR, the changing width (current research) is 73% larger for younger participants and 48% for older participants. An explanation about why this may be important from a clinical perspective is further described.

These differences in the ranges of the normative performance intervals are important for identifying motor control deficits on an individual basis. For example, a fixed performance interval for RT based on Grohs et al. [27] would only accurately identify deficits in that specific parameter for participants between 12 and 13 years old (where the changing range is closest to the fixed value). At 5 years old for RT, the performance interval is 64% wider than the fixed interval, and it is 49% smaller at 18 years old. At younger ages, the model would inaccurately identify participants who did not have impairment as having impairments, and for the older ages, it would inaccurately identify impaired performance as typical. The decreased size of the confidence intervals that result as a child reaches near-adult performance in the curves of typically developing participants better ensures that the model can accurately identify motor control deficits. These decreased confidence interval widths are most important for participants whose data is near the edges of the normative performance range.

While previous studies did not generate separate typical performance curves for male and female participants, some did generate separate curves for dominant and non-dominant hands [27,28]. Inter-sex differences in development have typically been found to be insignificant [39,40], so it is justifiable to create only one curve. Most other work did not account for sex [24]; however, the current work included the dummy-coded sex variable. Nemanich and Schildler-Ivens [41] used another device to examine bilateral coordination in children aged 7–17 years. They commented that “visual inspection of the plots and anecdotal observations during data collection suggested sex effects”, but found no significant effects when modeled. They argued that their sample size was too small to draw out these differences (29 participants). Given the early onset of puberty in females, they hypothesized that differences should be directly examined in future research. The addition of this variable to the current model gives the algorithm slightly more accurate differentiation between typical and atypical performance based on sex. The improved differentiation would only affect participants who were near the boundaries of the performance intervals, as the sex difference is often small relative to the magnitude of the performance parameter.

The importance of separating dominant and non-dominant hand performance in the typically developing population is not clear. There is contrasting evidence suggesting differences in performance [8,29,42]. The present study examines the ages from 5–18 and found that the movement area bias decreased by over 500% as the child reached near-adulthood. Scharoun and Bryden [43] suggest that there are three different phases of growth with respect to hand preference and motor skill refinement, from 3–5 years, 7–10 years, and 10–12 (near-adult performance). While the 7–10-year-olds prefer one hand over another, there is little difference in performance in speeded tasks. There is also evidence to suggest that lateralization of motor functions is more prominent in complex tasks of fine motor skills, unlike the tasks that were used for the present study [44]. Future work could compare the efficacy of both methods: separating data and generating different curves, or combining data and including the dummy-coded hand variable, and undertaking more complex tasks.

Recently, Hawe et al. conducted a study on characterizing complex upper limb movements in typically developing children, with and without visual aid [45]. They used a linear mixed-effects model to examine how age, hand (dominant vs. non-dominant), and visual feedback condition affected different aspects of movement [45]. The hand variable was also a dummy-coded variable (1 or 0), as was included in our model. In a similar manner to the sex variable, inclusion of the hand variable gives the algorithm slightly better differentiation between typical and atypical performance. Additionally, it allows easy comparisons between dominant and non-dominant hand performance for pathological populations. Separating the dominant and non-dominant hand is important for many different pathologies, including cerebral palsy and developmental coordination disorder. Previous studies creating normative models have either only used data from the dominant hand [46,47] or generated models for the dominant and non-dominant hand separately [27,28,31]. Since it is unclear if there are significant differences between dominant and non-dominant hand performance in typically developing populations, it is not clear whether it is necessary to generate separate curves for each hand. Using dummy-coded variables of the data from both hands in curve generation allows the doubling of the number of data points than if each hand were used separately. This doubling increases the repeatability of the curve fitting. The main drawback is the underlying assumption that both hands develop in the same manner.

### 4.1. Example Applications

As indicated, these normative models can be used to identify deficits in the performance of persons with different abilities. For example, Skarsgaard et al. [48] employed the Visually Guided Reaching (VGR) task with children who had developmental coordination disorder (DCD). A three-factor model was developed using exploratory factor analysis with the following performance parameters: Initial Direction Error, Minimum–Maximum Speed Difference, Path Length Ratio, Reaction Time, Movement Time, Maximum Speed, Initial Distance Ratio, and Speed Maxima Count. This model strongly differentiated the participants with DCD from the typically developing population of children.

Previous studies have required testing of age- and gender-matched control participants to allow for comparison to a pathologic population. For example, when examining bimanual coordination of children with unilateral cerebral palsy, Decraene found that bimanual coupling of the Object Hit (OH) task for children with cerebral palsy showed a preference (bias) toward the dominant hand, along with no age effects as compared to typically developing controls [49]. Use of a normative database might allow for greater differentiation than that of single-matched controls. A study by Khan et al. [50] also used the Kinarm for object-hit tasks concurrently with functional near infrared spectroscopy (fNIRS) to evaluate reaching accuracy and pre-frontal cortical activity. While matched participants are important for the fNIRS component, Kinarm performance of both groups could be compared to the normative population curves established within the present article.

Deficits in motor development could be identified using z-scores and the normative performance intervals from these models. After collecting data from a new participant, clinicians could calculate the z-score for each performance parameter for that participant using the normative curve and model of the residuals, as outlined in the Methods section. Deficits could be identified using a z-score cutoff, such as ±2, which corresponds to the 3rd and 97th percentiles, respectively. For example, if a participant’s z-score for reaction time were 2.4, they would be identified as having a deficit. Plotting new participant data onto the normative performance ranges shown in Figure 5 would also allow clinicians to quickly identify deficits through visual inspection. If the participant’s data point fell outside the normative range shown by the highlighted area, that would indicate a deficit.

### 4.2. Limitations

The main limitation of this work is the distribution of participant ages. While the participant group is large, there is a skew toward the older ages, and there are nearly twice as many male participants as female participants. These skewnesses are likely due to a recruitment bias. The most successful method of recruitment was partnering with an engineering summer camp for children and teens from 11 to 17 years old. This skew resulted in poorer confidence intervals on the curve fitting when compared to a uniform dataset. As such, the authors would suggest caution when using these curves to identify motor control deficits in children under 9 years old.

The existing skew in the collected dataset mostly affects the normative models at the ages below 9 years old, most prominently at 5 years old. The smaller sample size of participants at these younger ages makes it more likely that the models over- or under-estimate the speed of development at those ages. The exponential curves change most quickly at younger ages, so if the starting point of the curve (the predicted value of a 5-year-old’s performance) is higher than what is representative of the population, then it would look like performance improves faster with age than it actually would. Inaccuracies in the rate of development could result in misidentifying motor deficits or typical development trajectories in younger people.

Another limitation of this work is the choice to focus on chronological age rather than biological age. Biological age can be determined using longitudinal somatic assessments of growth, but the cross-sectional use of the peak height velocity (PHV) is a good predictor for a one-time prediction of biological age [51]. While biological age has been shown to affect motor performance in the typically developing population [26], especially in recent research, the authors did not incorporate the PHV evaluation in the present study. This is a possible direction for future study.

### 4.3. Future Work

Future work should involve recruiting more younger and more female typically developing participants. Recruiting and testing more young and female typically developing participants would improve the current models of typical development. Collaborating across multiple research institutes and groups could help facilitate this data collection. Recruiting and data collection efforts could be spread across a larger group, making it easier to reach the goal of having a more representative sample used for creating these normative models. When data collection started for this study, Kinarm did provide modified versions of the standard tasks for child participants, which made collaboration across research groups more difficult. Kinarm now provides modified standard tasks, which will facilitate collaboration on studies focused on child participants.

Additionally, the algorithm described could be used to create models of typical development based on other assessments. While the algorithm was used on data collected with the Kinarm robot, the same algorithm could be applied to data collected through different assessment tools.

## 5. Conclusions

The normative models provided could be used by clinicians to identify impairment, as well as to target and track therapies. The models can be used for robotic assessments in the same way typical performance intervals are used in existing clinical assessments, such as the Purdue Pegboard and Bruininks–Oseretsky Test of Motor Proficiency [13,14]. Clinicians could use the models to identify if a person has motor control deficits relative to others of the same age and sex. Therapies could then be targeted to the specific needs of the individual. These normative models could also be used to track the efficacy of therapies as they can record small improvements and large milestones, such as crossing the threshold into the normative ranges for different task parameters. The algorithm could be applied to other robotic assessments, with the Kinarm or other devices, to create normative ranges for measures of other aspects of function, such as position sense.

## Figures and Tables

**Figure 1 bioengineering-12-01240-f001:**
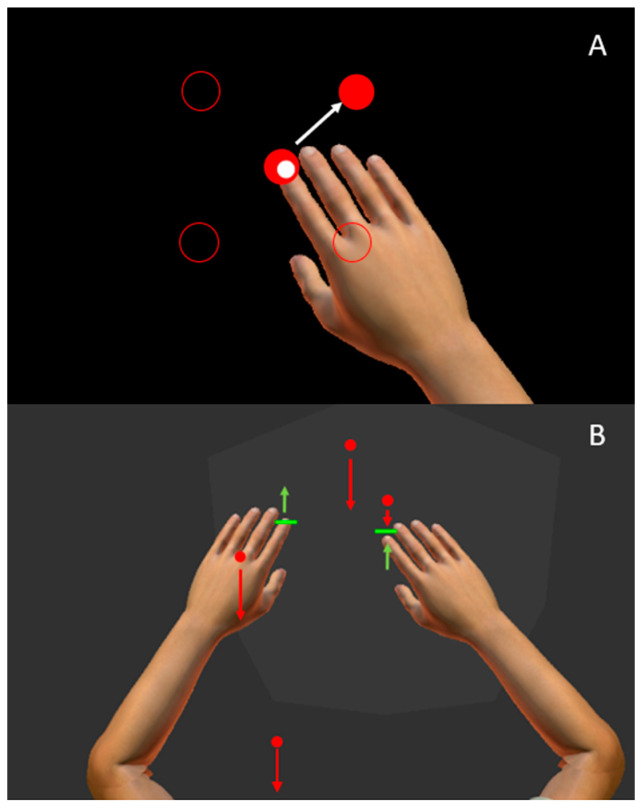
The Kinarm Exoskeleton Robot and representations of Visual Guided Reaching (VGR) and Object Hit (OH) tasks. (**A**) The participant’s finger is represented as a small white circle and positioned this white circle within a central red target. A peripheral target was then illuminated (upper right red circle), and participants then reached to that target. The positions of the other peripheral targets are represented by the red circle outlines. The white arrow indicates the direction a participant would reach. (**B**) a participant completing OH. Green paddles are aligned with the index finger of the participant. Participants were instructed to hit red balls that moved towards the participant at different speeds, represented by the length of the red arrows. The green arrows represent how a participant might move their hands to hit the red balls.

**Figure 2 bioengineering-12-01240-f002:**
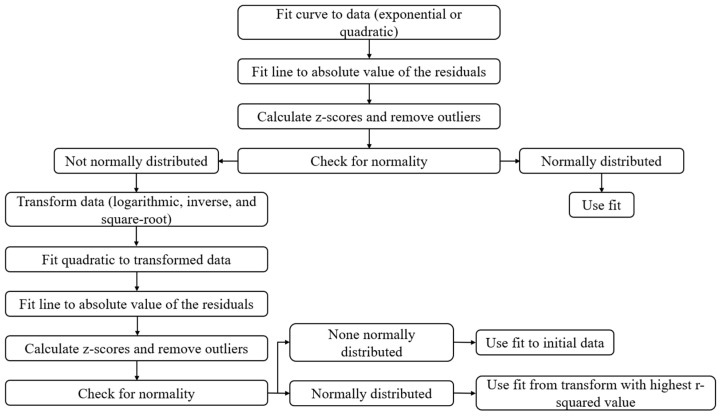
Schematic of the algorithm for creating normative models of each robotic task performance parameter.

**Figure 3 bioengineering-12-01240-f003:**
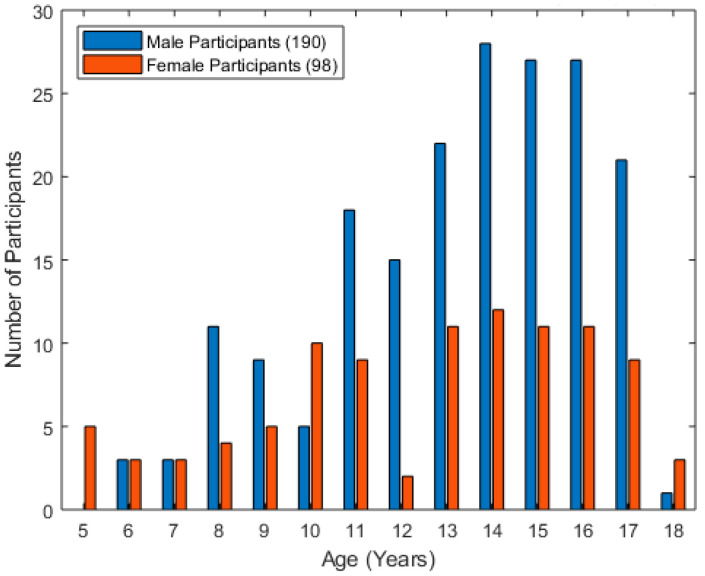
Histogram of the age of typically developing children who participated in the study.

**Figure 4 bioengineering-12-01240-f004:**
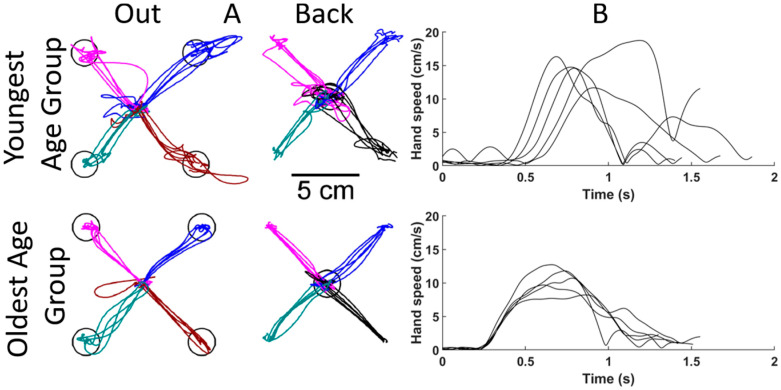
Comparisons of reaching for sex and handedness matched participants from the Visually Guided Reaching (VGR) task. The top row shows results from a younger participant (between 5 and 10 years old), and the bottom row from an older participant (between 13 and 18 years old). (**A**) Hand paths for each reach to each target. Each line represents an individual reaching trial, and the four colours represent reaches to/from each target. (**B**) Hand speeds for reaching out to the bottom right target. Each line represents the hand speed during an individual reaching trial.

**Figure 5 bioengineering-12-01240-f005:**
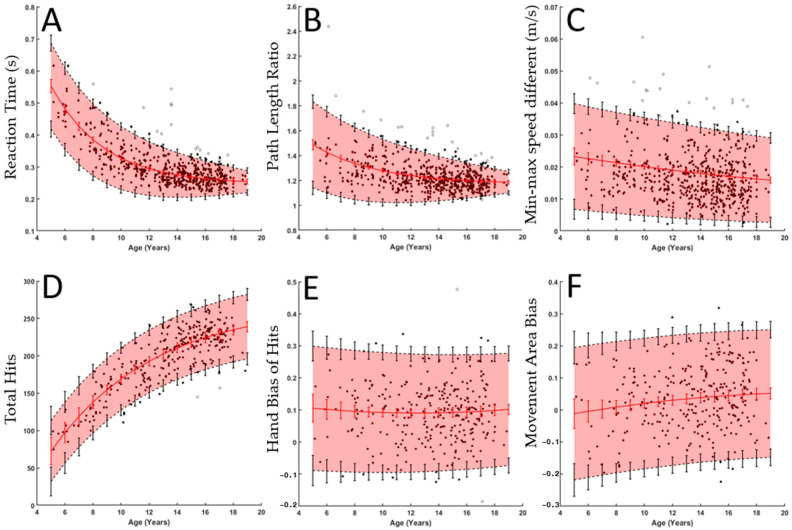
The top three panels represent raw data and fitted curves for three of the parameters from VGR ((**A**): Reaction Time, (**B**) Path Length Ratio, and (**C**) Min-Max Speed Difference), while the bottom panels represent parameters from Object Hit ((**D**) Total Hits, (**E**) Hand Bias of Hits, and (**F**) Movement Area Bias). In each subplot, black dots denote individual participants, and the red line and vertical red lines denote the fitted curve and its 95% confidence interval, respectively. Black dashed lines and vertical lines denote 2.5 and 97.5% performance, and the associated vertical lines denote a 95% confidence interval for each of these. The red shaded area is included to help visualize the central 95% range.

**Table 1 bioengineering-12-01240-t001:** Percentage changes in the mean curve and width of the 95% confidence intervals for all performance parameters.

Parameter Name	Percentage Change in Mean Value of Fitted Curve	Percentage Difference in the Width of the 95% Performance Confidence Intervals
Visually Guided Reaching		
Posture Speed	−40	−57
Reaction Time	−54	−73
Initial Direction Angle	−44	−63
Initial Distance Ratio	17	−50
Initial Speed Ratio	5	−71
Speed Maxima Count	−18	−17
Movement Time	−29	−7.5
Path Length Ratio	−20	−72
Max Speed	25	28
Min-Max Speed Difference	−31	−18
Object Hit		
Total Hits	229	6
Median Error	30	75
Miss Bias	0	69
Hand Bias of Hits	−4	−10
Hits (Dominant)	198	35
Hits (Non-Dominant)	216	39
Hand Speed (Dominant)	−10	−51
Hand Speed (Non-Dominant)	−4	−49
Hand Speed Bias	−42	−7
Movement Area (Dominant)	38	11
Movement Area (Non-Dominant)	−1	−7
Movement Area Bias	−533	−3

## Data Availability

The original contributions presented in the study are included in the article/Appendix A, further inquiries can be directed to the corresponding author.

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
