# Peer review of "Non-Linear Modeling of Motor Development in Typically Developing Children and Youth Aged 5–18 Years Using Robot-Based Behavioral Assessments"

_bioengineering, 2025, doi:10.3390/bioengineering12111240_

Round 1
Reviewer 1 Report (New Reviewer)
Comments and Suggestions for Authors
Recommendation: Minor Revision.
It is a well-considered and significant work that explores the nonlinear motor development of typically developing children aged 5-18 years through KINARM robotic tasks. The authors' compilation of extensive, well-organized pediatric data represents a substantial achievement. The rigorous modelling methodology and the inclusion of age-related variability mark important progress for both research and practical applications.
To strengthen the paper further:
1. Clarify the novelty (Introduction, lines 73-89): briefly describe how this study advances beyond Grohs et al. and Hawe et al., particularly by modelling changing variability and accounting for sex and hand effects.
2. Balance participant demographics (Methods 95-107; Results 254-258): Include a brief age-sex table or histogram and discuss how this may influence results, especially in the youngest age groups.
3. Clarify methods (198-253): add a short overview or schematic to simplify understanding of the modelling process.
4. In the discussion (335-376): mention biological explanations for lower variability, such as neural system maturation or improved feedforward regulation.
5. Application (425-444): Give a brief example of how z-scores could be utilized to identify abnormal motor development.
Minor adjustments to characters, consistent use of vocabulary (e.g., 'visually guided reaching'), and polished language would enhance an already strong paper. Overall, it is a high-quality, impactful study likely to inspire further research in pediatric motor control.
Sincerely.
Comments on the Quality of English LanguageThe manuscript was very clear and well written. The English is good in general and conveys the scientific material well. Even minimal light edits to the language would help in making the paper even easier to read.
You can look into making some of the longer sentences shorter and sometimes write in the active voice to make the literature a little more straightforward and interesting. Naming would also be beneficial—e.g., Visually Guided Reaching (VGR) and Object Hit (OH) must be typed in a similar manner everywhere.
Do not ignore minor points, either, like spelling rules (the journal is committed to American English, or, in any case, "modeling," "behavioral," etc.), the usage of hyphens in the compounds (it is always used in the journal in case of compounds), and trivial typographical errors (e.g., to hit red balls, not to high).
All these are little modifications. Overall, the manuscript has good readability, and a short revision of the ethical editorial would render an already good paper even more readable and official.
Author Response
Please see the attachment.

Reviewer 2 Report (New Reviewer)
Comments and Suggestions for Authors
This manuscript presents a technically strong and conceptually valuable study that models motor development in developing children and adolescents (ages 5–18). The study is well-structured, the rationale is sound, and the results are clearly presented. The modelling strategy particularly the inclusion of age-related variability is a notable advance over previous literature.
However, the sample composition remains a significant limitation. While the total sample (n = 288) is larger than in most comparable studies, it is not evenly distributed across ages, with a heavy skew toward older participants and nearly twice as many males as females. This heterogeneity restricts the generalisability of the normative curves, especially for the younger age range (<9 years). The authors note this issue but should more explicitly situate their dataset relative to what an ideal normative database would require; that is a more balanced representation across sex, age, and biological maturity, with sufficient participants per age band to achieve stable curve estimation. My view is that rather than publish now, that this limitation to be addressed.
The paper would also benefit from a clearer acknowledgement of the practical and logistical challenges inherent in collecting such data (e.g., access to specialised robotic equipment, testing time, recruitment constraints). Framing this study as an important first step toward building a collaborative or multi-centre normative database would strengthen its contribution and realism.
In its current form, the study provides valuable preliminary models but does not yet achieve the level of representativeness required for definitive normative reference standards. The authors are encouraged to act on their own future research recommendations, expand the dataset (particularly among younger and female participants), and resubmit once these data are available. With those enhancements, the manuscript could become a key reference for developmental motor robotics and clinical benchmarking.
Comments on the Quality of English LanguageGenerally fine.
Two-hundred and eighty-eight participants who are typically developing (ages 5-18) completed a robotic point-to-point reaching task and an object hitting task using the Kinarm Exoskel- 19 thephrase typically developing is awkward.
eton
Round 2
Reviewer 2 Report (New Reviewer)
Comments and Suggestions for Authors
I appreciate the way you have addressed the earlier comments and improved the paper in several important areas.
You have strengthened the manuscript by: a) Providing a much clearer acknowledgment of the dataset limitations, including the uneven age and sex distribution; b) Adding reference to the simulation study validating the reliability of curve fitting for ages nine and above, c) Including a thoughtful discussion of the logistical and practical challenges of recruiting younger participants and using specialised equipment. However, I think there is a need to say that if the case for doing the research is valuable, the effort to collect the data is justifiable. d) Explaining past barriers to collaboration and how new task standardisations now make future multicentre work feasible.
You are doing the the appropriate action by framing the current study appropriately as a foundational step toward a broader normative database. These changes make the discussion more transparent and realistic, and they help readers better understand both the value and the constraints of your findings.
Revisions
One area that could be developed further, either briefly in this version or in future work, is a clearer statement about how the sampling imbalance may affect the interpretation of developmental trends. This would make the discussion of limitations even more robust.
Overall, this is a clear and well-revised manuscript, and the revisions have meaningfully improved its scientific and practical contribution.
Comments on the Quality of English LanguageThe English is readable.
Author Response
Please see the attachment.

This manuscript is a resubmission of an earlier submission. The following is a list of the peer review reports and author responses from that submission.
Round 1
Reviewer 1 Report
Comments and Suggestions for Authors
Dear authors,
Firstly, I would like to congratulate you for this paper. The topic it addresses is highly interesting. Secondly, from a humble point of view, I share some considerations so as to improve it:
In title: In order to be more specific, I suggest that authors could indicate the age range of participants instead of “children and youth”.
In abstract: Indicate explicitly the aim of the research.
In keywords: In order to increase the visibility in the different databases this journal is indexed in, try to avoid repeating keywords from those which already are part of the title.
In introduction: Review journal’s guidelines with regard to references. Several of them are repeated (1 an3, 2 and 4). Provide a more detailed background about previous empirical results of other studies.
In material and methods: Split it into these subsections: a) participants, b) measurements, c) procedures and d) Data analysis.
In results: Delete line 211 (3.1. Subsection).
In discussion: When possible, try to update the references, especially from 2024-2025. Provide more potential future works.
In conclusion: This section is adequate.
In references: Review the capital letters (e.g., S. C. Dobri in line 361 but S. C. D. Dobri in line 375).
Reviewer 2 Report
Comments and Suggestions for Authors
Dear Authors,
thank you for a chance of reviewing this study.
This is generally a solid piece of scholarly work, yet, there are some shortages that make this manuscript not ready for publishing at this moment.
The idea of the study is certainly original, but needs more grounded rationale and a reader needs to understand what has been done in this area of robot-based assessment so far in terms of motor development.
Below, please find some suggestions and comments that will hopefully help you enhance the quality of the paper.
Title is informative but suggests a broader study on motor development than just a one-characteristic (RT) from within the broad scope of motor development characteristics. I think, this may be misleading and needs to be adjusted to the context of your study.
Key words are fine.
Introduction - this section requires more work. First of all, it barely touches the essence of the problem - the development of nervous system. In case of the parameter you had selected for the study on motor development - reaction time - it is mainly about the development of the nervous system. This section should include some paragraphs on trajectory of biological maturation with clear differentiation between early school-age children (school readiness, right/handedness, lateralization) and adolescents. I see some issues with the reliability of your findings and their generalization - you don't mention anything whether you have asked about other contradicting and possibly mitigating factors like sport experience or computer gaming, ect. which are typical for this age category of youth.
These factors change the quality of performance a lot, especially in adolescents practicing combat sports (with a potential bias from for example a interhemispheric transfer as a result of other, out of school activities) or in changes in the nervous system and brains of teenagers spending extensive hours on gaming.
For example, if one practiced fencing or table tennis with right hand for 7 years, it would obviously be visible in their results in RT, accuracy and precision timing with dominant hand but not with the other one, ect. The same with gaming.
In the Introduction section you also need to provide a solid rationale why you had chosen this parameter (RT) - your ownership of the technology is not enough, your choice needs to be academically justified.
Material and methods section is the longest section of the paper - perhaps it is better to divide it into subsections - Participants, Methods, Tools, Ethics, ect. - but I leave it up to the Authors and Journal Editors.
Results -this is generally clear section, graphs of the figures are neat and findings are clearly described, the only questions concerns 3.1. subsection that is missing?
Discussion - this section also requires strengthening. You have to associate your findings with neuro-motor developmental trajectories, and discuss how reaction time changes with age and maturation. But there are also other issues, for example - you repeat in the text the information about the uniqueness of your sample size as compared to the other study, you question the need for separating performance accordingly to handedness - dig in the lateralization literature and sport-specialization as well to bring some valid arguments to the table.
Conclusions need to be re-written with the limitation that they concern just one characteristic of motor development, although you could suggest how other parameters might be included in similar robot-based behavior assessment.
Reference are up-to-date and relevant for the study, but I am sure with broadening your Introduction and Discussion this section will be extended.
Good luck with your study.
Reviewer 3 Report
Comments and Suggestions for Authors
The work is of little scientific value. Due to the demonstrated limitations, the research results cannot be implemented in practice. The manuscript represents the level of conference proceedings, not a study for a high-impact journal. Main drawback: the selection of the research group and its description.
Detailed comments:
- Introduction: too short, cursory, and a brief review of the literature on the subject of the research (17 lines). The tasks on lines 47-48 are unnecessary; this is not the purpose of the research.
- Introduction, research purpose (line 45) "large cohort" – this does not provide information about the research subject (large–small).
- Introduction, research purpose (line 46) "typically participants." What does "typically participants" mean?
- Materials and Methods: the selection of the research group and its description are incorrectly described. Anthropometric information is missing.
- Materials and Methods (line 50). What does "typically developing participants" mean? How was this assessed?
- Materials and Methods (line 58) "or had any physical or cognitive impairment that would affect the completion of the study" – meaning what impairments?
- Materials and Methods. No justification for the tests used. Why were these particular tests chosen?
- Conclusions: The statements in the conclusions contradict each other when we compare the Limitation and Conclusion. Compare Limitation (line 318) "authors would suggest caution..." and the statement in Conclusions (line 322) "could be used..."—lack of consistency. The limitations identified by the authors and my comments prove that the scientific value of the research described in the manuscript is low.
- References: Items 6, 13, 14, 15, 20, and 21 are from several decades ago. Their inclusion in the manuscript requires justification. There is likely much more current literature on the subject.
- Mess in the text: line 211, "3.1. Subsection".
Round 2
Reviewer 1 Report
Comments and Suggestions for Authors
Dear authors,
I would like to congratulate you for your effort. This new version of the papar is, in my humble view, much better than the previous one.
Author Response
Thank you very much for reviewing the article. Your input was invaluable for improving our work, and we appreciate all the effort you put in to give us such detailed feedback.
Reviewer 2 Report
Comments and Suggestions for Authors
Dear Authors
I see you have incorporated the comments into the text of your article and it is more sound now and thorough. And concerning reaction time remark and your response - I know that it was only one of the parameters, but it was most striking and the one that the reader quickly can link with other studies and other findings when looking for some analogies - my pointing at the shortages in this area also should give you an idea that the other parameters were not discussed in full depth, but now after the revision you have introduced I see the paper readers better and it is more coherent.
You have not only made a substantial changes to the text, but also extended references - all together it was for the better for the article.
I have no further comments.
Author Response
Thank you very much for your detailed feedback. Your input was invaluable to help us improve the article. We appreciate all the time and effort you put in to help us improve our work.
Reviewer 3 Report
Comments and Suggestions for Authors
Despite several changes suggested by the reviewer (review No. 1), the manuscript still does not meet the requirements of a study of a high scientific standard.
The work is of little scientific value. Due to the demonstrated limitations, the research results cannot be implemented in practice. The manuscript represents the level of conference proceedings or a journal with a small Impact Factor, not a study for a high-impact journal.
Main drawback: the selection of the research group and its description. It's very general. It does not characterize the group of people studied. Simply reporting the age of the subjects is not enough. It would probably also be worthwhile to consider the differences between chronological age and biological age. It's common knowledge that there are significant developmental differences between children and adolescents.
In the manuscript, the authors did not provide the date of obtaining consent to conduct the research or the year in which the research was conducted. Only the consent number was provided (the Health Sciences and Affiliated Hospitals Research Ethics Board (HSREB) at Queen’s University, Kingston, Ontario, Canada (application number 6004951). High-level research standards require such data.
The authors did not address the fundamental comment regarding the description of the examined persons (participants). The authors did not answer all the reviewer's questions.
A response like "Reply: We added a paragraph..." is not an answer. It's not the reviewer's role to search the manuscript for an answer (paragraph). Authors should provide precise information in their response (e.g., questions 1, 6?). This is standard practice. The responses are cluttered and lack numbering consistent with the reviewer's questions (see point 6).
Regarding some of the comments (e.g., 4, 6?, 8), I will not engage in further discussion with the authors, as their response is inappropriate (rude). The authors' example of the 1905 article is inappropriate towards the reviewer.
For a scientist, it's obvious what anthropometric data are. It's obvious to a scientist that they don't need to justify citations in the manuscript (main text), but rather convince the reviewer that a given publication should have been cited.
